# Impact of *Lactobacillus acidophilus* and Its Combination with Isoflavone Products on Calcium Status, Calcium Transporters, and Bone Metabolism Biomarkers in a Post-Menopausal Osteoporotic Rat Model

**DOI:** 10.3390/nu16152524

**Published:** 2024-08-02

**Authors:** Iskandar Azmy Harahap, Marcin Schmidt, Ewa Pruszyńska-Oszmałek, Maciej Sassek, Joanna Suliburska

**Affiliations:** 1Department of Human Nutrition and Dietetics, Faculty of Food Science and Nutrition, Poznan University of Life Sciences, 60-624 Poznan, Poland; 2Department of Biotechnology and Food Microbiology, Faculty of Food Science and Nutrition, Poznan University of Life Sciences, 60-637 Poznan, Poland; 3Department of Animal Physiology, Biochemistry and Biostructure, Faculty of Veterinary Medicine and Animal Science, Poznan University of Life Sciences, 60-637 Poznan, Poland

**Keywords:** isoflavones, probiotics, calcium, bone health, postmenopausal, osteoporosis

## Abstract

Osteoporosis in menopausal women requires alternatives to current medications, considering their adverse effects. In this context, probiotics and isoflavone products are promising dietary interventions. The objective of our study was to examine the impacts of *Lactobacillus acidophilus* and its combination with daidzein and tempeh on calcium status, calcium transporters, and bone metabolism biomarkers in a post-menopausal osteoporotic rat model. A total of 48 female Wistar rats were exposed to a two-stage experiment involving calcium deficit induction and subsequent dietary interventions across six groups. Calcium levels, the gene expression of TRPV5 and TRPV6 calcium transporters, bone histopathology, serum bone metabolism markers, and blood biochemistry were evaluated. The results revealed that, while decreasing serum calcium levels, the groups that received the probiotic *L. acidophilus* and isoflavone combination exhibited increased bone metabolism biomarkers and decreased calcium transporter expressions, akin to the effects of bisphosphonate. Additionally, significant improvements in bone histopathology were observed in these groups. However, the group receiving probiotic *L. acidophilus* alone did not exhibit significant changes in bone resorption biomarkers, calcium transporter expression, or various blood parameters. Meanwhile, the combination of probiotic *L. acidophilus* with tempeh positively influenced hematological parameters and reduced cholesterol and triglyceride levels, but it led to elevated blood glucose levels. Correlation analyses highlighted associations between serum calcium levels, calcium transporter expression, and bone metabolism biomarkers. In conclusion, our findings suggest that the daily consumption of probiotic *L. acidophilus* in combination with isoflavone products may improve bone health in ovariectomized rats, warranting further research to elucidate potential interactions with other nutrients.

## 1. Introduction

Probiotics are live micro-organisms that offer health benefits when consumed in adequate amounts [1]. Research has indicated that specific probiotic strains, such as *Lactobacillus acidophilus*, can influence the composition of the gut microbiota [2] and improve nutrient absorption [3], including calcium [4], which may contribute to better bone health. *L. acidophilus*, known for its positive effects on gastrointestinal health, has gained attention for its potential role in promoting bone health through the modulation of the gut microbiota. While the exact mechanisms by which *L. acidophilus* affects bone health are still being studied, it is believed that they may impact bone homeostasis through various pathways, such as regulating inflammatory cytokines, improving nutrient absorption, and influencing immune responses that indirectly affect bone remodeling processes [5]. Moreover, an asymmetry in the composition of the gut microbiota, referred to as dysbiosis, has been associated with a range of health issues, including osteoporosis. The idea of using microbiome-based therapies for common human diseases is fascinating [6]. The concept of the gut–bone axis highlights the intricate relationship between the gut microbiota and bone metabolism [7], suggesting that microbial-derived metabolites and signaling molecules produced in the gut can affect bone remodeling processes through systemic circulation or local interactions with bone cells. Understanding how probiotics, gut dysbiosis, and the gut–bone axis interact is crucial for understanding their potential therapeutic applications in preventing and managing osteoporosis.

However, combining probiotics with isoflavones has emerged as a promising dietary component for enhancing calcium levels and supporting bone health [8]. These bioactive compounds, notably found in soybeans and soy-based products, have received considerable attention due to their potential benefits for bone metabolism. Isoflavones possess phytoestrogenic properties that mimic the effects of estrogen in the body, which can positively impact bone mineral density [9]. Together, probiotics and isoflavones present a promising nutritional strategy for preventing osteoporosis and reducing the risk of fractures.

Isoflavones, classified as phytoestrogens, are plant-derived compounds that structurally resemble estrogen and exhibit estrogenic effects in the body [10,11,12]. These compounds have gained significant attention due to their potential role in improving calcium balance and bone health. Mechanistically, isoflavones influence bone metabolism through various pathways, including regulation of the RANKL/RANK/OPG system and the modulation of osteoblast and osteoclast activity [13]. Rich sources of isoflavones include fermented soybean products such as tempeh, which undergo microbial fermentation processes that enhance the bioavailability and activity of these compounds [14,15]. In addition to tempeh, the effects of daidzein—a prominent isoflavone found in soy products—are particularly known due to its potential health benefits in post-menopausal women. Daidzein acts as a weak estrogen agonist and is believed to contribute to the protective effects of soy against osteoporosis. The unique chemical composition of isoflavones, combined with their ability to mimic estrogenic effects, position them as promising dietary agents for preserving bone health and reducing the risk of osteoporosis. Understanding the mechanisms underlying the bone-protective effects of isoflavones and their natural dietary sources is crucial for elucidating their therapeutic potential in managing skeletal disorders.

A comprehensive understanding of bone metabolism relies on assessing various biomarkers that reflect the processes of bone resorption and formation. Pyridinoline (PYD) and deoxypyridinoline (DPD) are well-established biomarkers of bone resorption, reflecting the breakdown of collagen fibers in bone tissue. Similarly, *C*-telopeptide of type I collagen (CTX) serves as a reliable indicator of bone resorption activity, providing insights into the rate of bone turnover [16]. Conversely, biomarkers of bone formation, such as Bone Alkaline Phosphatase (BALP), Osteocalcin (OC), and Procollagen Type I *N*-Terminal Propeptide (PINP), offer valuable information regarding the processes involved in the synthesis and mineralization of new bone tissue [17]. These biomarkers serve as proxies for the activity of osteoblasts—the cells responsible for bone formation—and provide crucial insights into bone health and remodeling dynamics.

In addition to these biomarkers, calcium transporters play a pivotal role in maintaining calcium homeostasis, a fundamental aspect of bone metabolism. TRPV5 and TRPV6 are key calcium transporters involved in this process. These calcium channels, found in epithelial cells, exhibit a notable preference for calcium ions (Ca^2+^). Research has indicated that these channels play a vital role in maintaining the body’s calcium equilibrium through facilitating the absorption of Ca^2+^ in the intestines and the re-absorption of Ca^2+^ in the kidneys. Dysregulation of this process has been implicated in conditions such as osteoporosis and calcium metabolism disorders. Furthermore, TRPV channels modulate osteoblast and osteoclast differentiation, which are pivotal processes in bone remodeling. Notably, TRPV5/TRPV6 channels have been identified on the surface of osteoclasts, where TRPV5 functions as a negative regulator, modulating bone resorption triggered by RANKL signaling to maintain bone homeostasis [18]. Therefore, investigating the expression and regulation of TRPV5 and TRPV6 provides valuable insights into the mechanisms underlying calcium homeostasis and its impact on bone physiology.

Our previous investigation in a healthy female rat model yielded promising findings regarding calcium status [19], calcium transporter expression, and bone metabolism biomarkers [20]. However, a notable research gap exists regarding the effects of probiotics and their combination with isoflavone products on these parameters in a post-menopausal osteoporotic context. Despite the fact that our previous study provided valuable insights into the baseline physiology of bone health, it is imperative to evaluate the impacts of dietary interventions—particularly those involving the combination of probiotics and isoflavone products—on bone health. In our previous study, we investigated the impact of tempeh and daidzein on calcium metabolism and bone biomarkers in ovariectomized rats [21]. Expanding upon these findings, the current study explores the effects of *L. acidophilus* and its combination with isoflavone products (including tempeh and daidzein) on calcium status, calcium transporters, and bone metabolism biomarkers in a rat model of post-menopausal osteoporosis. Incorporating our previous research outcomes, the objective of the present study was designed to bridge this gap through evaluating the influence of a combined intervention involving probiotic *L. acidophilus* combined with isoflavones on calcium status, calcium transporter expression, and bone metabolism biomarkers in a rat model of post-menopausal osteoporosis. Thus, the purpose of this study was to evaluate the effects of the combination of the probiotic *L. acidophilus* and isoflavones on calcium status, calcium transporter expression, and bone metabolic biomarkers in an ovariectomized rat model. Additionally, we aimed to assess the impact of this combination on various hematological parameters in order to obtain a comprehensive understanding of its effects on overall health. For this study, we selected *L. acidophilus* DSM20079 due to its well-documented probiotic properties, such as surviving gastrointestinal transit, adhering to intestinal epithelial cells, and modulating immune responses [22,23], which are essential for enhancing calcium absorption and improving bone metabolism. This strain has been extensively studied for its safety and efficacy [24], making it a reliable choice for investigating the potential benefits in post-menopausal osteoporotic rat models. We hypothesized that this intervention would positively influence these parameters, potentially offering a novel therapeutic approach for managing post-menopausal osteoporosis. Through this investigation, we expected to elucidate the effects of these dietary components on bone health in menopausal individuals, providing valuable insights for the development of novel osteoporosis management strategies.

## 2. Materials and Methods

### 2.1. Materials and Ethical Considerations

This study used 48 female Wistar rats, aged 3 months, obtained from the Nencki Institute of Experimental Biology (Warsaw, Poland). The preparation of tempeh and probiotic powder followed the methodology described in our previous work [20]. The AIN 93M diet, Augusta variety soybeans, pure daidzein, alendronate sodium trihydrate, calcium citrate tetrahydrate, and other chemicals used have been detailed in our prior study [21].

Ethical approval was granted by the Local Ethical Committee in Poznań, Poland (registration number 21/2021, 21 May 2021). This study adhered to national and international guidelines, including the NIH Guide for the Care and Use of Laboratory Animals, Directive 2010/63/EU, and relevant Polish legislation, with procedures conducted in compliance with the ARRIVE guidelines.

### 2.2. Animal Housing and Surgical Procedures

Female Wistar rats, aged 3 months, were housed in a controlled environment at the Department of Human Nutrition and Dietetics, Poznań University of Life Sciences, Poland. They were kept at 21 ± 2 °C with 55–65% relative humidity, on a 12 h light/dark cycle, in pairs within stainless steel cages. The rats were acclimatized for 1 week with ad libitum access to Labofeed B and tap water.

Following acclimatization, the rats were divided into two groups: a sham operation group (S, *n* = 8) and a bilateral ovariectomy group (OVX, *n* = 40). Sham operations served as controls to compare against the ovariectomy group. The inclusion of a sham control group in this study was aimed at simulating a procedure or treatment experience without the actual application of the procedure or test substance [25]. All surgeries were performed under anesthesia with ketamine and Cepetor, adhering to sterile techniques. Post-surgery, rats were placed on a heated mat for recovery and monitored for distress, with veterinary care available as needed. A 7-day observation period followed, with rats receiving a semi-synthetic diet (AIN-93M) [26] and unrestricted access to tap water.

Following a 7-day recovery, the initial body weights of the rats were measured to ensure proper randomization. The 40 OVX rats were then randomly assigned to five groups of eight based on body weight, which is crucial for reducing bias and ensuring group comparability [27]. Previous research supports that eight rats per group provides adequate statistical power to detect significant effects [28].

To evaluate the effects of *L. acidophilus* and isoflavone products on bone health, we fed the rats a low-calcium diet to simulate post-menopausal conditions. This method helped to assess the therapeutic potential of our interventions against osteoporosis risks due to decreased estrogen and calcium levels. Table 1 outlines the dietary composition during this period. In our previous study [21], we detailed the composition of the AIN 93M diet. The calcium-to-phosphorus (Ca:P) ratio was calculated based on available data, with a ratio of 2.51 g/g as provided by Zoolab (Sędziszów, Poland). The calcium-deficient diet was provided for 3 weeks (stage 1), followed by standard and modified diets for 6 weeks (stage 2). Group 1 included sham rats (S, *n* = 8) on a standard diet, and Group 2 consisted of OVX rats (O, *n* = 40) on a calcium-deficient diet. Daily dietary intake was monitored, with deionized water provided ad libitum. A three-week period is sufficient to induce calcium deficiency [29].

Following the induction of calcium deficiency, both the S and OVX groups were switched to a standard diet containing calcium citrate tetrahydrate as the source of calcium. The OVX group was further divided into five groups: O group fed with AIN 93M; OB group fed with AIN 93M and bisphosphonate; OL group fed with AIN 93M and probiotic *L. acidophilus*; ODL group fed with AIN 93M, daidzein, and probiotic *L. acidophilus*; and OTL group fed with AIN 93M, tempeh, and *L. acidophilus*. The specific dietary formulations are detailed in Table 2. During the 6-week intervention stage, the diets and deionized water were available to the rats without restriction.

In this study, the doses of daidzein and tempeh were meticulously selected based on their respective isoflavone contents. We conducted a laboratory analysis to determine the isoflavone concentration in the tempeh sample through its preparation. Our analysis revealed that 250 g of tempeh corresponded to an equivalent of 10 mg of daidzein. This information guided our dosage selection, ensuring that appropriate levels of daidzein and tempeh were chosen. To ensure uniformity, the quantity of pure daidzein was adjusted to match the concentration specified in 250 g of tempeh. To incorporate 250 g/kg of tempeh flour and 10 mg/kg of daidzein into the AIN93M diets, we carefully modified the diets by replacing starch with 250 g and 10 mg of the relevant components. This method ensured that the diets included the appropriate amount of tempeh flour or daidzein while maintaining nutritional consistency. We calculated the isoflavone dosages based on previous research demonstrating their efficacy in improving bone health in both mice and humans [30]. Administering alendronate bisphosphonate at a dosage of 3 mg/kg/day over an extended period of time promoted bone remodeling and facilitated the healing of fractures in rodents that underwent ovariectomy [31]. We used rat body weight measurements to determine weekly dose adjustments for alendronate bisphosphonate.

The selection of probiotic dosage was informed by prior research demonstrating its beneficial impacts on both rodent and human bone health. Specifically, we referred to the study by Dar et al., who investigated ovariectomized female mice fed diets containing *L. acidophilus* with a daily dose of 10^9^ CFU/day over 6 weeks [32]. The authors observed a decrease in osteoclastogenic factor expression and an increase in antiosteoclastogenic factor expression with the 10^9^ CFU/day dose; however, a more significant effect was noted in postmenopausal women receiving a higher probiotic dose (10^10^ CFU/day), compared to a lower dose (2.5 × 10^9^ CFU/day) [33]. Therefore, we chose to administer *L. acidophilus* at a dosage of 10^10^ CFU/day in our investigation. Figure 1 provides a flowchart depicting the research process from adaptation to intervention periods.

### 2.3. Monitoring of Body Weight, Dietary Intake, and Decapitating the Rats

During the intervention stage, rats in each group were weighed weekly using a calibrated scale (RADWAG PS 750.X2, Radom, Poland). Meticulous records of daily food consumption were maintained for each group throughout the experiment. Rats received a daily allocation of fresh food and deionized water, with any leftovers from the previous day promptly removed to maintain uninterrupted access to nourishment. This standardized protocol ensured consistent food and water availability, which is crucial for preserving the well-being of rats and facilitating their normal growth [34]. The food efficiency ratio (FER) reflects the efficiency with which animals convert consumed food into body weight, which is calculated by dividing the weight gained by the amount of food consumed during the study period. Moreover, providing fresh food and water daily and removing any remnants effectively prevented spoilage, ensuring that the rats consumed a fresh and untainted diet and water throughout the study. Three days before the end of the intervention period (i.e., three days before euthanasia), all rats in each experimental group underwent body composition analysis using Bruker’s All New 2nd Generation Minispec LF90II Body Composition Analyzer, MA, USA. This assessment enabled the quantification of fat mass (in grams).

After the intervention stage, rats underwent a fasting period lasting 4–6 h before their body weight was measured using a calibrated scale. This fasting protocol aimed to mitigate any potential influence of recent food intake on subsequent weight measurements. Following the weight assessment, euthanasia was carried out via decapitation—a well-established and ethically acceptable method in experimental animal research. Decapitation ensures swift and painless termination, thereby minimizing the likelihood of distress.

### 2.4. Collection of Blood, Serum, Bone, and Feces

Blood samples were collected to analyze blood morphology. The serum was transferred to sterilized tubes and left to clot at room temperature for 30 min. Afterward, the samples were centrifuged at 4 °C for 15 min at 2000 rpm to separate the blood cells. Femoral bone samples were meticulously extracted, removing surrounding tissue. Fecal samples were gathered from each rat’s cage. All samples were stored at −80 °C until analysis.

### 2.5. Analysis of Blood Morphology and Biochemistry Parameters and Femoral Bone Histopathology

Whole-blood morphological and biochemistry parameters, as well as femoral bone histopathology assessments, were conducted at Alab Laboratories in Poznań, Poland. Erythrocytes, hemoglobin, hematocrit, mean corpuscular volume (MCV), mean corpuscular hemoglobin (MCH), mean corpuscular hemoglobin concentration (MCHC), platelets, red cell distribution width–coefficient of variation (RDW-CV), leukocytes, neutrophils, lymphocytes, monocytes, eosinophils, basophils percentage, alanine aminotransferase (ALT), aspartate aminotransferase (AST), cholesterol, glucose, and triglycerides were measured using standard laboratory techniques. The measurements were performed according to established protocols using automated analyzers and commercially available assay kits. The units of measurement for each parameter were as follows: erythrocytes (Y/L), hemoglobin (g/dL), hematocrit (%), MCV (fL), MCH (pg), MCHC (g/dL), platelets (G/L), RDW-CV (%), leukocytes (G/L), neutrophils (G/L), lymphocytes (G/L), monocytes (G/L), eosinophils (G/L), basophils percentage, ALT (U/L), AST (U/L), cholesterol (mg/dL), glucose (mg/dL), and triglycerides (mg/dL). For bone histopathology assessment, femoral bone samples underwent rigorous preparation, including evaluation of fixation upon arrival, followed by a 14 h de-calcification in EDTA solution and subsequent immersion in 70% ethanol for over 24 h. After confirming the adequacy of preparation, trimmed sections were sealed in labeled histology cassettes and processed using standard paraffin techniques. Histological staining with hematoxylin–eosin was performed to visualize cellular and tissue structures. Pathological evaluations were carried out by experienced veterinarian pathologists using Zeiss Axiolab 5 microscopes in Halle, Germany at magnifications of 5×, 10×, and 40×. Grading of histopathological changes and morphometric analysis of trabecular bone were conducted using established criteria and protocols. Representative areas were captured using a 3DHISTECH PANNORAMIC 250 Flash III microscope, Budapest, Hungary, and digital slides were generated with the Grundium Ocus^®^20 microscope slide scanner, Tampere, Finland, ensuring meticulous documentation of findings.

### 2.6. Analysis of Calcium Concentration

Calcium concentrations in diets and fecal samples were determined by ashing 2 g of each diet in a muffle furnace at 450 °C until complete mineralization, followed by dissolution in 1 N nitric acid (Suprapure, Merck). Meanwhile, calcium levels in femoral bone were assessed after digestion in 65% (*w*/*w*) spectra pure HNO_3_ (Merck, Kenilworth, NJ, USA) using a Microwave Digestion system (Speedwave Xpert, Berghof, Eningen, Germany). Calcium concentrations in diet, serum, fecal, and bone samples were determined via flame atomic absorption spectrometry (AAS-3, Carl Zeiss, Jena, Germany) after dilution with Lanthanum (III) chloride (Merck KGaA, Darmstadt, Germany) and deionized water. The calcium content was analyzed at a wavelength of 422.7 nm. The accuracy and dependability of the method were assessed using bovine liver 1577C (Sigma-Aldrich, St. Louis, MO, USA) as a certified reference material. The study of this reference material yielded data that showed a high level of accuracy in the method used. Specifically, the accuracy rate for quantifying calcium was calculated to be 92%.

### 2.7. Analysis of Bone Metabolism Biomarkers

ELISA kits were acquired from Qayee Bio-Technology Co., Ltd. (Shanghai, China) for the purpose of measuring serum concentrations of bone metabolism indicators. The ELISA kits were utilized in conjunction with the LEDetect96 absorption spectrophotometry instrument from Labexim in Lengau, Austria. Pyridinoline (PYD), deoxypyridinoline (DPD), and *C*-telopeptide of type I collagen (CTX) were specifically quantified as biomarkers to assess bone resorption. Conversely, Bone Alkaline Phosphatase (BALP), Osteocalcin (OC), and Procollagen Type I *N*-Terminal Propeptide (PINP) were measured as biomarkers to evaluate bone formation.

### 2.8. Analysis of TRPV5 and TRPV6 Calcium Transporters

Quantitative real-time polymerase chain reaction (qRT-PCR) was utilized to evaluate the expression of calcium transporters, following the method outlined in our previous study [21]. The primers used were: Gapdh (forward: TGACTTCAACAGCGACACCCA, reverse: CACCCTGTTGCTGTAGCCAAA), TRPV5 (forward: CGAGGATTCCAGATGC, reverse: GACCATAGCCATTAGCC), and TRPV6 (forward: GCACCTTCGAGCTGTTCC, reverse: CAGTGAGTGTCGCCCATC).

### 2.9. Statistical Analysis

The variables were evaluated for normal distribution using the Shapiro–Wilk method. The statistical significance of the detected differences was assessed using analysis of variance (ANOVA) and Tukey’s post hoc test. A significance level of 5% was used to determine if there were significant differences between the groups. Moreover, it was estimated that a statistical power of 80% could be attained when detecting significance at the 0.05 level by utilizing a sample size of 8 rodents in each group. Pearson’s correlation analysis was conducted to evaluate the associations between serum calcium levels, bone metabolism indicators, and calcium transporters. The statistical analysis and creation of figures were conducted using SPSS version 22 for Windows. The data are reported as mean values along with their respective standard deviations.

## 3. Results

Comparing the calcium deficit (AIN_CaDef) diet to the standard diet with calcium (S & O), no significant differences were noted in their food energy calories, while differences were observed for calcium content (Table 3). The AIN_CaDef group in stage 1 exhibited a significantly lower calcium content compared to the other diet groups in stage 2. No significant differences in calcium content were observed among the S & O, OB, OL, ODL, and OTL diets.

### 3.1. Body Weight Gain, Body Fat Mass, and Food Intake

Table 4 presents the findings related to body weight gain, body fat mass, food intake, and calcium intake among the experimental rat groups throughout the study. In the comparison between the S and O groups, during stage 1 (aimed at inducing a calcium deficit), no significant differences in body weight gain were observed across all rat groups. However, in the final body weight, the O group had a significantly higher body weight compared to the S group. Moreover, the percentage of FER in the O group was significantly higher than in the S group. No significant differences were observed in fat mass, food intake, and calcium intake between the S and O groups.

In the comparison between the O group and the groups receiving modified diets (OB, OL, ODL, and OTL), no significant differences were found in body weight gain during both stage 1 and final body weight, fat mass, and food intake. However, the OTL group exhibited a significantly higher calcium intake, compared to the S and O groups.

### 3.2. Impact on Blood Morphological and Biochemical Parameters

Table 5 outlines the blood morphology profiles observed in rats following the 6-week intervention with modified diets. When comparing the S and O groups, the O group exhibited significantly elevated levels of blood parameters including leukocytes, neutrophils, lymphocytes, eosinophils, and cholesterol.

In the comparison between the O group and the groups receiving modified diets (OB, OL, ODL, and OTL), the ODL group showed a significant increase in eosinophil levels. Conversely, the OT group displayed a significant decrease in cholesterol levels, compared to the O group. Notably, when compared to the S group, the OTL group had higher hemoglobin and hematocrit levels. Furthermore, the OL and OTL groups exhibited significantly higher glucose levels, and the OTL group showed significantly lower triglyceride levels compared to the S group.

### 3.3. Impact on Calcium Status and Bone Metabolism Biomarkers

Table 6 presents the serum calcium, fecal calcium, and bone metabolism biomarkers measured in rats fed with modified diets. A notable decrease in the serum calcium level was observed in the O group, compared to the S group. However, no significant differences were noted for calcium levels in bone and fecal samples, as well as bone metabolism biomarkers, between the S and O groups.

Comparing the O group with the groups receiving modified diets (OB, OL, ODL, and OTL), the OL, ODL, and OTL groups showed significant decreases in serum calcium levels. Conversely, the OB, OL, ODL, and OTL groups displayed significant increases in calcium levels in the femoral bone compared to the O group.

Regarding bone metabolism biomarkers, the OL, ODL, and OTL groups exhibited significant increases in PYD and CTX levels compared to the S group. In addition, the OL group showed a significant increase in PYD levels compared with the O group. The OL, ODL, and OTL groups demonstrated significant increases in OC and PINP levels compared to the S and O groups. Furthermore, the OTL group showed a significant increase in DPD levels compared to the S and O groups, whereas the OB group presented a significant increase in BALP levels compared to the O group.

### 3.4. Impact on Histopathological Changes in Femoral Bone

The histopathological changes depicted in Figure 2 illustrate the significant differences between the experimental groups. Specifically, the O group showed an increased presence of medullary spaces, characterized by a larger surface area occupied by adipocytes and a corresponding area devoid of bone marrow components, indicating an osteoporotic condition compared to the S group. In contrast, the ovariectomized groups fed with different intervention diets (OL, ODL, and OTL) showed similar effects to the ovariectomized group treated with bisphosphonate (OB), demonstrating a reduction in the surface area occupied by adipocytes within the femoral bone structure.

### 3.5. Impact on Calcium Transporters

Figure 3 presents the mRNA expression levels of calcium transporters (TRPV5 and TRPV6) in the duodenum and jejunum. Comparison between the S and O groups did not reveal any significant differences in the mRNA expression of TRPV5 and TRPV6 in either the duodenum or jejunum. However, comparison of the O group with the groups receiving modified diets (OB, OL, ODL, and OTL) revealed a significant decrease in the mRNA expression of TRPV5 in the duodenum for OB, OL, ODL, and OTL. Additionally, a significant reduction in the mRNA expression of TRPV5 was observed in the jejunum of the OTL group, compared to the S group.

### 3.6. Correlation between Calcium Status, Calcium Transporters, and Bone Metabolism Biomarkers

In Figure 4, the Pearson’s correlation analysis results are illustrated, revealing the relationships among calcium status, calcium transporters, and bone metabolism biomarkers. Figure 4A shows significant positive correlations, notably between serum calcium levels and TRPV5 expression in the duodenum (r = 0.475). Conversely, Figure 4B demonstrates prominent negative correlations, particularly between serum calcium levels and Procollagen Type I *N*-Terminal Propeptide (r = −0.748). Figure 4C,D highlight the correlations between calcium transporters and bone metabolism biomarkers, indicating a significant positive correlation between TRPV6 expression in the duodenum and deoxypyridinoline (r = 0.365), as well as a prominent negative correlation between TRPV5 expression in the duodenum and Procollagen Type I *N*-Terminal Propeptide (r = −0.386).

## 4. Discussion

In contrast to our previous studies in healthy female rats, where daily intake of pure daidzein and probiotic *L. acidophilus* did not significantly increase calcium levels in femoral bones [19] or sera [20], our current study in ovariectomized rats showed that the daily intake of a combination of probiotic *L. acidophilus* and tempeh decreased calcium levels in serum while increasing them in femoral bone. This suggests that isoflavones—especially daidzein, which is found in fermented soy foods such as tempeh—may play a role in restoring calcium balance and reducing the risk of developing post-menopausal osteoporosis. Through their estrogenic activity, these isoflavones may act as partial substitutes for declining endogenous estrogen levels, potentially regulating bone turnover and calcium homeostasis [35,36]. Additionally, the relationship between *L. acidophilus* and calcium absorption [37] may contribute to the mechanism underlying the decreasing calcium levels in sera. These findings suggest that the combination of probiotic *L. acidophilus* and isoflavone products modulates calcium homeostasis, potentially affecting skeletal calcium levels.

A key finding of this study was that daily consumption of probiotic *L. acidophilus* and its combination with isoflavone products led to a significant increase in femoral bone calcium levels, accompanied by a reduction in serum calcium levels. This decrease in serum calcium levels could be attributed to calcium re-distribution from the bloodstream to the bone, resulting in increased calcium levels within the femoral bone. This phenomenon of calcium deposition in bone leading to a decrease in serum calcium levels is in line with previous research, indicating that interventions with probiotics and isoflavones promoting bone health can induce such calcium re-distribution [19]. Therefore, the observed decrease in serum calcium levels could potentially indicate augmented calcium deposition in bone. However, at present, this remains an observational finding without a definitive mechanism elucidated at the molecular level. Further investigations are warranted to delineate the underlying mechanisms by which *L. acidophilus* and its combination with isoflavone products influence bone metabolism and calcium status in post-menopausal osteoporotic rats.

Furthermore, consuming a combination of probiotic *L. acidophilus* and isoflavone products may have beneficial effects on both bone formation and resorption metabolism after menopause, potentially improving overall bone health. Isoflavones, as phytoestrogens found in soy products such as tempeh, have been shown to exert estrogen-like effects on bone cells, promoting osteoblast activity and inhibiting osteoclast-mediated bone resorption [38]. Additionally, probiotics such as *L. acidophilus* may influence taurine metabolism and the composition of the intestinal microbiota. Taurine—an amino sulfonic acid—is involved in modulating calcium signaling and is primarily biosynthesized in the liver. Its antioxidant properties can also improve gastric injury [39]. These combined effects may have contributed to the observed improvements in bone metabolism biomarkers among the intervention groups.

The purpose of analyzing the expression of calcium transporters—specifically, TRPV5 and TRPV6—was to understand how our dietary interventions might influence calcium absorption in post-menopausal osteoporotic rats. TRPV5 and TRPV6 are epithelial Ca^2+^ channels known for their high selectivity for calcium absorption. Our findings indicate that bisphosphonates, probiotic *L. acidophilus*, and the latter’s combination with isoflavone products down-regulated TRPV5 expression in the duodenum and jejunum. This suggests a potential role of these interventions in modulating calcium absorption in post-menopausal osteoporotic rats. However, it is important to interpret these changes in mRNA expression cautiously, as they may not always correlate directly with protein activity [40].

One notable correlation we observed was a significant positive association between serum calcium levels and the expression of TRPV5 in the duodenum. This correlation aligns with our current findings: the intake of isoflavones and probiotic *L. acidophilus* simultaneously decreased calcium transporter expression and serum calcium levels. The decrease in serum calcium levels and the reduction in calcium transporter expression may result from complex regulatory mechanisms aimed at maintaining calcium homeostasis. This alteration to the calcium transport mechanisms might occur in response to changes in dietary intake, hormonal signals, or other factors to maintain overall calcium balance within cells and tissues. For instance, the parathyroid gland plays a crucial role in regulating calcium levels; primarily through parathyroid hormone, which stimulates bone resorption, enhances intestinal calcium absorption, and increases active renal calcium absorption [41]. Additionally, we observed a significant positive correlation between TRPV6 expression in the duodenum and DPD levels, suggesting that bone resorption may increase as calcium absorption decreases.

Conversely, the significant negative correlation between PINP and serum calcium levels, as well as TRPV5 expression in the duodenum, suggests an inverse relationship between calcium distribution and bone formation in ovariectomized rats. This finding aligns with our current study, where the intake of isoflavone products and probiotic *L. acidophilus* simultaneously decreased serum calcium levels and TRPV5 expression, followed by an increase in PINP concentrations. This negative correlation may imply that increased TRPV5 expression and enhanced serum calcium levels are associated with impaired bone formation, resulting in lower levels of PINP.

In addition to these findings on calcium status and calcium transporter expression, the daily consumption of probiotic *L. acidophilus* and tempeh simultaneously resulted in significantly higher levels of hemoglobin, hematocrit, and glucose, along with significantly lower levels of cholesterol and triglycerides. Another notable finding in our study is the observed increase in leukocytes and lymphocytes among ovariectomized rats. This increase may be attributed to dysregulation of the immune system associated with estrogen deficiency resulting from ovariectomy. Estrogen plays a crucial role in modulating immune function, including regulating the production and activity of various immune cells [42]. Therefore, the removal of estrogen through ovariectomy may lead to an imbalance in immune cell populations, resulting in an increase in leukocytes and lymphocytes as part of the body’s response to the hormonal changes associated with menopause.

In previous studies, probiotics such as *Lactobacillus* species have shown varying effects on hemoglobin and hematocrit levels. While our previous research demonstrated no significant changes with probiotic *L. acidophilus* alone or in combination with daidzein and genistein in healthy female rats [43], our current findings suggest that daily consumption of probiotic *L. acidophilus* and tempeh may enhance hemoglobin and hematocrit levels. This effect could be attributed to mechanisms such as improved iron bioavailability facilitated by *L. acidophilus* and the potential erythropoietic peptides generated during tempeh fermentation [44,45,46].

In addition, favorable outcomes were observed in terms of lipid metabolism, with a decrease in cholesterol and triglyceride levels following the daily intake of probiotic *L. acidophilus* and its combination with tempeh. These findings are consistent with previous research demonstrating the potential of dietary interventions—such as consuming tempeh and co-incubating probiotics during soy tempeh fermentation—to improve lipid profiles in diabetic rat models [47,48]. However, it is worth noting that an adverse effect on blood glucose levels was also observed, with increases observed in the OL and OTL groups. Despite this, previous studies have shown that daily consumption of tempeh led to improvements in blood glucose levels in diabetic rat models, possibly due to the presence of short-chain fatty acids, such as propionate, which enhance the uptake of glucose by skeletal muscles and regulate glucose levels through the release of GLP-1. Additionally, propionate inhibits liver gluconeogenesis and modulates lipid metabolism, thereby reducing blood glucose levels. Alterations in the gut microbiota composition—particularly, reductions in propionate producers—can lead to elevated glucose levels, highlighting the relationships between diet, the gut microbiota, and metabolic health [49]. Furthermore, the enhancement of isoflavone absorption mediated by lactic acid bacteria, such as *Lactobacillus*, suggests a potential mechanism for improving overall metabolic health [50]. Thus, while this study revealed some adverse effects on glucose levels, the observed benefits in lipid metabolism underscore the potential effects of dietary interventions involving probiotic *L. acidophilus* and its combination with tempeh.

Our findings suggest that *L. acidophilus* intake may contribute to reduced blood cholesterol and lipid levels through mechanisms involving the down-regulation of microsomal fatty acid elongation and glycerolipid metabolism pathways, as well as the inhibition of key cholesterol biosynthesis-related enzymes by soy isoflavones [51,52]. These systemic effects—although not the primary focus of our study—underscore the potential implications of our interventions beyond bone health. Future research is needed to elucidate the interconnected pathways linking glucose–lipid metabolism and bone health.

### Limitations and Future Perspective of Study

In our study, several strategies were implemented to enhance the robustness and reliability of the findings. First, the inclusion of a sham group in the study design allowed for the control of potential confounding factors associated with the surgical procedure, thus ensuring that observed effects were specifically attributable to the interventions under investigation. Additionally, the incorporation of a group treated with the drug bisphosphonate provided a valuable reference point for comparing the efficacy of the interventions against a conventional pharmacological approach commonly used in osteoporosis management. This comparative analysis with a conventional osteoporosis management drug offers valuable context for the research findings, offering insights into the potential benefits of utilizing *L. acidophilus* and its combination with isoflavone products in post-menopausal osteoporotic rats.

While our findings revealed promising trends in calcium metabolism and bone biomarkers, several limitations should be acknowledged. Our study did not include measurements of parathyroid hormone (PTH)—a crucial parameter in calcium metabolism—which limits our ability to comprehensively evaluate calcium homeostasis. Furthermore, although our research suggested that the combination of *L. acidophilus* with isoflavone products (e.g., tempeh and daidzein) might offer synergistic effects on calcium status and bone health, the role of taurine in calcium signaling and bone metabolism was not investigated. Future studies exploring taurine levels and its interactions with other biomolecules could provide valuable insights into new therapeutic approaches for osteoporosis.

Considering the growing significance of environmentally friendly approaches for handling bone health in the elderly demographic, our discoveries encourage nutritional support in the context of pharmacological treatment. Future research endeavors should explore optimal dosages, durations, and modes of administration for isoflavones and probiotics in order to maximize their therapeutic effects. Additionally, investigating potential synergistic effects with lifestyle modifications such as exercise could enhance bone health outcomes. Ultimately, the translation of our findings to clinical practice warrants human clinical studies to validate the efficacy and safety of the interventions in this population. To further enhance the depth of our research findings, future studies could incorporate the methodology to measure BMD and BMC using DEXA, in addition to analyzing bone metabolism biomarkers. This approach would allow for a comprehensive assessment of changes in bone architecture over time. Through including DEXA measurements, future research endeavors could identify any gradual changes in bone structure, providing valuable insights into the efficacy of interventions studied.

## 5. Conclusions

In conclusion, the daily consumption of probiotic *L. acidophilus* and its combination with isoflavone products may enhance femoral bone calcium levels while concurrently reducing serum calcium levels in ovariectomized rats. Additionally, this intervention also led to promising improvements in bone metabolism biomarkers, with comparable effects to bisphosphonate on bone histopathology. Furthermore, consumption of probiotic *L. acidophilus* and tempeh simultaneously appeared to positively influence hematological parameters and reduce cholesterol and triglyceride levels. In addition, the daily intake of probiotic *L. acidophilus* alone or in combination with tempeh resulted in elevated blood glucose levels. However, it is worth noting that the daily intake of probiotic *L. acidophilus* alone did not significantly affect bone resorption biomarkers, calcium transporter expression, or various blood parameters in ovariectomized rats, highlighting the need for further research to elucidate the potential interactions between probiotics and other nutrients in modulating bone health.

## Figures and Tables

**Figure 1 nutrients-16-02524-f001:**
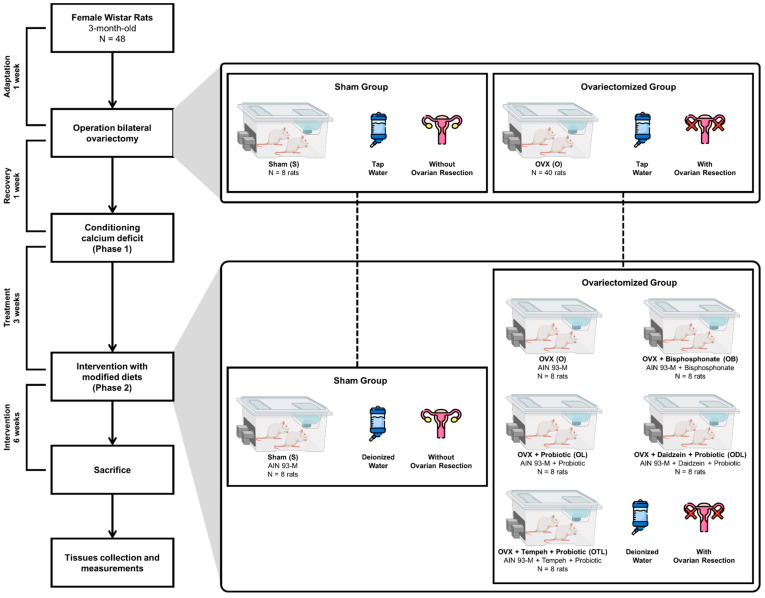
Experimental design from adaptation to intervention periods. AIN 93M: a formulated diet that contains essential nutrients for rodents; OVX: ovariectomized rats.

**Figure 2 nutrients-16-02524-f002:**
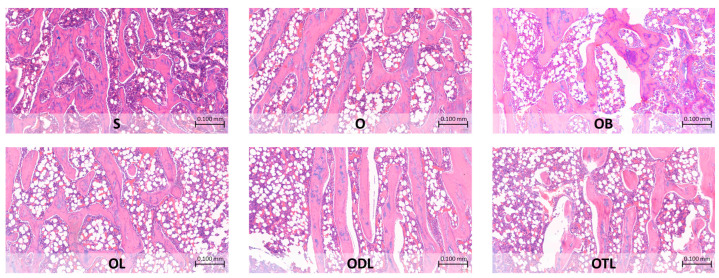
Histopathological changes in femoral bone after a 6-week treatment with dietary modifications. S: sham rats fed AIN 93M; O: ovariectomized rats fed AIN 93M; OB: ovariectomized rats fed AIN 93M with bisphosphonate; OL: ovariectomized rats fed AIN 93M with probiotic; ODL: ovariectomized rats fed AIN 93M with daidzein and probiotic; OTL: ovariectomized rats fed AIN 93M with tempeh and probiotic. Photos were taken under objective 10× and a scale of 0.100 mm.

**Figure 3 nutrients-16-02524-f003:**
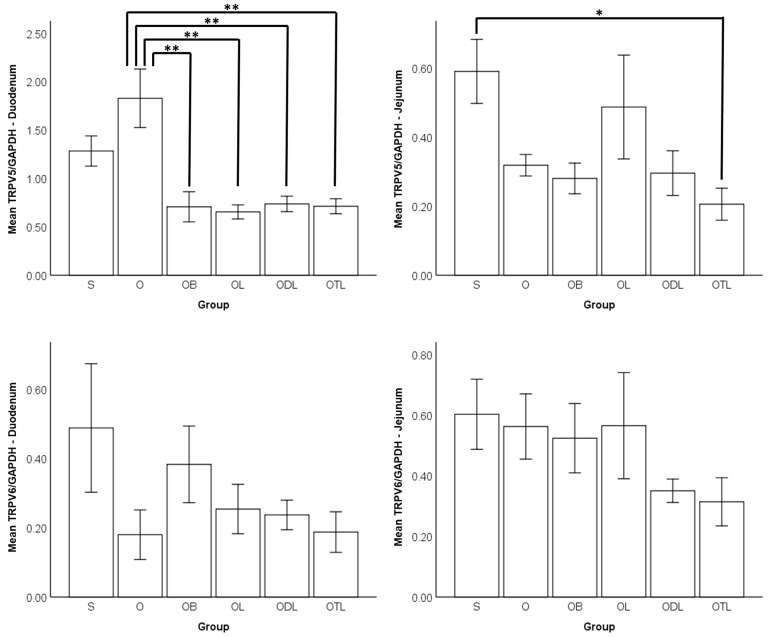
mRNA expression of the calcium transporters TRPV5 and TRPV6 in the duodenum and jejunum. TRPV5: Transient Receptor Potential channel family Vanilloid subgroup 5; TRPV6: Transient Receptor Potential channel family Vanilloid subgroup 6. S: sham rats fed AIN 93M; O: ovariectomized rats fed AIN 93M; OB: ovariectomized rats fed AIN 93M with bisphosphonate; OL: ovariectomized rats fed AIN 93M with probiotic; ODL: ovariectomized rats fed AIN 93M with daidzein and probiotic; OTL: ovariectomized rats fed AIN 93M with tempeh and probiotic. Values (means ± SD) are presented as cycle threshold values for gene expression analysis and as the relative abundance of TRPV5 and TRPV6 proteins normalized to the reference protein GAPDH. Results of ANOVA followed by Tukey’s post hoc honestly significant difference test showing significant differences between types of diet. Data are presented as mean ± standard deviation. *: statistically significant difference (*p* < 0.05) compared to the control group (S). **: statistically significant difference (*p* < 0.05) compared to the control group (O).

**Figure 4 nutrients-16-02524-f004:**
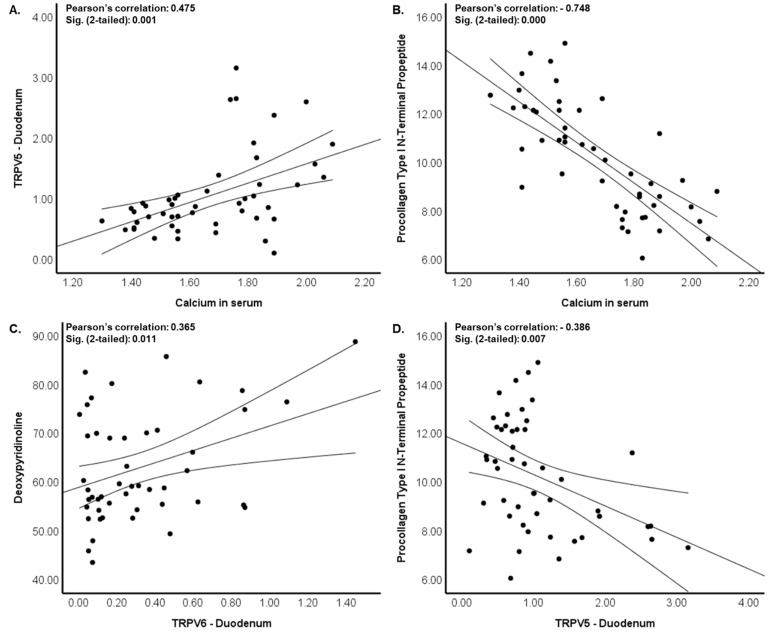
Pearson’s correlation between calcium status, calcium transporters, and bone metabolism biomarkers. (**A**) A significant positive correlation between serum calcium levels and TRPV5 expression in the duodenum; (**B**) A significant negative correlation between serum calcium levels and Procollagen Type I N-Terminal Propeptide; (**C**) A significant positive correlation between TRPV6 expression in the duodenum and deoxypyridinoline; (**D**) A significant negative correlation between TRPV5 expression in the duodenum and Procollagen Type I N-Terminal Propeptide. TRPV5: Transient Receptor Potential channel family Vanilloid subgroup 5; TRPV6: Transient Receptor Potential channel family Vanilloid subgroup 6.

**Table 1 nutrients-16-02524-t001:** Dietary formulas used during the calcium deficiency period.

Code	Group	Number of Rats	Composition
S	Sham	8	AIN 93M
O	OVX	40	AIN 93M with calcium deficit

AIN 93M: A formulated diet that contains essential nutrients for rodents; OVX: ovariectomized rats. The standard diet was based on the AIN-93M formulation [26], which was designed to meet the nutritional requirements of adult rodents.

**Table 2 nutrients-16-02524-t002:** Dietary formulas used during the intervention with modified diets period.

Code	Group	Number of Rats	Composition
S	Sham	8	AIN 93M
O	OVX	8	AIN 93M
OB	OVX + Bisphosphonate	8	AIN 93M + Bisphosphonate
OL	OVX + Probiotic	8	AIN 93M + *L. acidophilus*
ODL	OVX + Daidzein + Probiotic	8	AIN 93M + Daidzein + *L. acidophilus*
OTL	OVX + Tempeh + Probiotic	8	AIN 93M + Tempeh + *L. acidophilus*

AIN 93M: A formulated diet that contains essential nutrients for rodents; OVX: ovariectomized rats.

**Table 3 nutrients-16-02524-t003:** Energy and calcium contents in diets.

Parameter	Type of Diet					
	AIN_CaDef	S & O	OB	OL	ODL	OTL
	Stage 1	Stage 2	Stage 2	Stage 2	Stage 2	Stage 2
Energy (Kcal/g dry mass)	3947.78 ± 551.38	3883.34 ± 573.92	3850.51 ± 586.98	3936.88 ± 481.91	3924.61 ± 526.70	3865.97 ± 759.89
Calcium (mg/g dry mass)	0.02 ± 0.01 *	5.06 ± 0.30	4.66 ± 0.71	4.87 ± 0.47	5.34 ± 0.54	6.06 ± 0.88

AIN_CaDef: an AIN 93M diet without calcium content; S & O: sham and ovariectomized rats fed AIN 93M; OB: ovariectomized rats fed AIN 93M with bisphosphonate; OL: ovariectomized rats fed AIN 93M with probiotic; ODL: ovariectomized rats fed AIN 93M with daidzein and probiotic; OTL: ovariectomized rats fed AIN 93M with tempeh and probiotic. Stage 1: calcium deficiency treatment; Stage 2: treatment with dietary modifications. Data are presented as mean ± standard deviation. The composition of diets in AIN_CaDef, S & O, and OB groups were as previously described in our paper [21]. The composition diet in the intervention groups was as follows: protein content (mg/g dry mass) = OB: 133.12 ± 3.59, OL: 112.62 ± 0.67, ODL: 122.37 ± 1.45, and OTL: 196.24 ± 0.4; fiber content (mg/g dry mass) = OB: 39.48 ± 1.46, OL: 32.40 ± 0.71, ODL: 34.86 ± 1.28, and OTL: 43.57 ± 0.51; fat content (mg/g dry mass) = OB: 41.93 ± 0.26, OL: 35.23 ± 1.53, ODL: 39.18 ± 0.30, and OTL: 68.53 ± 0.55; and carbohydrate content (mg/g dry mass) = OB: 715.44 ± 144.47, OL: 776.14 ± 124.95, ODL: 753.23 ± 128.91, and OTL: 594.28 ± 188.57. * indicates statistically significant differences (*p* < 0.05) compared to the control group (AIN 93M, S & O groups).

**Table 4 nutrients-16-02524-t004:** Final body weight, body weight gain, and fat mass in the rats.

Parameter	Group					
	S	O	OB	OL	ODL	OTL
Initial body weight (g)	275.88 ± 18.79	294.50 ± 20.63	292.38 ± 20.32	294.50 ± 20.14	294.63 ± 20.52	294.63 ± 20.89
Body weight gain in stage 1 (g)	22.38 ± 13.06	33.00 ± 21.84	39.13 ± 16.50	38.00 ± 9.40	31.63 ± 7.76	32.00 ± 12.63
Final body weight (g)	305.50 ± 31.51	355.38 ± 23.54	346.75 ± 28.41	352.00 ± 27.92 *	340.00 ± 36.77	344.50 ± 33.50
Final fat mass (g)	62.94 ± 29.63	86.95 ± 14.93	89.53 ± 24.69	77.83 ± 19.19	75.88 ± 35.24	56.51 ± 26.54
Food intake (g/day)	16.75 ± 1.10	17.78 ± 1.14	17.27 ± 0.96	17.54 ± 1.31	16.93 ± 1.10	16.67 ± 1.38
FER (%)	43.27 ± 47.08	156.76 ± 95.50	88.33 ± 62.97	111.19 ± 58.15	81.21 ± 87.19	107.24 ± 53.99
Calcium intake (mg/day)	85.28 ± 5.60	90.52 ± 5.79	80.96 ± 4.52	85.86 ± 6.42	90.99 ± 5.90	101.68 ± 8.42 **

S: sham rats fed AIN 93M; O: ovariectomized rats fed AIN 93M; OB: ovariectomized rats fed AIN 93M with bisphosphonate; OL: ovariectomized rats fed AIN 93M with probiotic; ODL: ovariectomized rats fed AIN 93M with daidzein and probiotic; OTL: ovariectomized rats fed AIN 93M with tempeh and probiotic. Body weight gain was calculated as the difference in body weight between the end and the beginning of each stage. FER: food efficiency ratio (weight gain (g)/food intake (g) × 100). Results of ANOVA analysis followed by Tukey’s post hoc honestly significant difference test showing significant differences between types of diet. Data are presented as mean ± standard deviation. * indicates statistically significant difference (*p* < 0.05) compared to the control group (S). ** indicates statistically significant difference (*p* < 0.05) compared to both control groups (S & O).

**Table 5 nutrients-16-02524-t005:** Blood morphological and biochemical parameters in rats fed modified diets.

Parameter	Group					
	S	O	OB	OL	ODL	OTL
Erythrocytes (Y/L)	7.94 ± 0.40	8.18 ± 0.32	8.09 ± 0.33	8.01 ± 0.40	8.26 ± 0.30	8.41 ± 0.39
Hemoglobin (g/dL)	14.99 ± 0.65	15.25 ± 0.41	15.44 ± 0.50	15.29 ± 0.60	15.59 ± 0.30	15.98 ± 0.55 *
Hematocrit (%)	42.80 ± 1.51	43.55 ± 1.30	43.75 ± 1.87	42.44 ± 1.81	44.65 ± 0.74	45.34 ± 2.55 *
MCV (fL)	53.98 ± 1.61	53.40 ± 1.83	54.20 ± 1.24	53.05 ± 2.79	54.03 ± 1.71	53.96 ± 1.24
MCH (pg)	18.90 ± 0.96	18.69 ± 0.51	19.15 ± 0.67	19.13 ± 1.13	18.85 ± 0.81	19.03 ± 0.64
MCHC (g/dL)	35.01 ± 1.01	35.03 ± 0.65	35.31 ± 1.10	36.03 ± 0.63	34.90 ± 0.71	35.26 ± 1.06
Platelets (G/L)	836.25 ± 74.53	838.75 ± 123.39	861.88 ± 144.46	798.00 ± 82.32	732.50 ± 115.44	801.25 ± 70.78
RDW-CV (%)	12.60 ± 0.50	12.83 ± 0.42	12.83 ± 0.71	13.16 ± 0.65	12.74 ± 0.52	12.81 ± 0.34
Leukocytes (G/L)	7.34 ± 2.20	13.65 ± 2.79	13.91 ± 2.10 *	16.38 ± 4.43 *	13.35 ± 2.91 *	15.68 ± 4.89 *
Neutrophils (G/L)	0.98 ± 0.28	1.79 ± 0.58	1.88 ± 0.53 *	1.85 ± 0.54 *	1.46 ± 0.43	1.49 ± 0.31
Lymphocytes (G/L)	5.45 ± 1.96	10.67 ± 2.44	10.61 ± 1.82 *	12.98 ± 4.46 *	10.36 ± 2.46 *	13.01 ± 4.76 *
Monocytes (G/L)	0.70 ± 0.25	0.85 ± 0.32	1.00 ± 0.44	1.11 ± 0.36	1.09 ± 0.38	0.85 ± 0.26
Eosinophils (G/L)	0.20 ± 0.05	0.30 ± 0.05	0.38 ± 0.12 *	0.39 ± 0.06 *	0.41 ± 0.08 ***	0.31 ± 0.06 *
Basophils %	0.45 ± 0.15	0.29 ± 0.10	0.26 ± 0.09	0.33 ± 0.10	0.38 ± 0.10	0.26 ± 0.11 *
ALT (U/L)	37.02 ± 4.96	44.44 ± 8.23	45.79 ± 6.66	44.79 ± 13.45	46.33 ± 6.63	49.63 ± 12.31
AST (U/L)	155.01 ± 41.32	184.36 ± 40.92	167.86 ± 41.36	175.48 ± 86.04	180.98 ± 67.96	212.27 ± 112.37
Cholesterol (mg/dL)	81.03 ± 20.85	106.51 ± 12.24	110.08 ± 16.40 *	115.41 ± 17.16 *	106.45 ± 16.08 *	80.67 ± 14.02 **
Glucose (mg/dL)	115.99 ± 12.96	132.30 ± 15.53	132.59 ± 15.53	135.42 ± 8.77 *	131.59 ± 12.08	140.43 ± 11.20 *
Triglycerides (mg/dL)	210.70 ± 100.72	166.03 ± 71.82	164.48 ± 73.72	205.28 ± 114.05	125.47 ± 52.97	89.63 ± 30.74 *

S: sham rats fed AIN 93M; O: ovariectomized rats fed AIN 93M; OB: ovariectomized rats fed AIN 93M with bisphosphonate; OL: ovariectomized rats fed AIN 93M with probiotic; ODL: ovariectomized rats fed AIN 93M with daidzein and probiotic; OTL: ovariectomized rats fed AIN 93M with tempeh and probiotic. Results of ANOVA analysis followed by Tukey’s post hoc honestly significant difference test showing significant differences between types of diet. Data are presented as mean ± standard deviation. * indicates statistically significant difference (*p* < 0.05) compared to the control group (S). ** indicates statistically significant difference (*p* < 0.05) compared to the control group (O). *** indicates statistically significant difference (*p* < 0.05) compared to both control groups (S & O).

**Table 6 nutrients-16-02524-t006:** Calcium status and bone metabolism biomarkers.

Parameter	Group					
	S	O	OB	OL	ODL	OTL
Calcium in serum (mmol/L)	1.94 ± 0.13	1.80 ± 0.05	1.82 ± 0.07 *	1.55 ± 0.10 ***	1.53 ± 0.02 ***	1.42 ± 0.03 ***
Calcium in femoral bone (mg/g dry mass)	239.51 ± 15.32	212.13 ± 76.82	354.31 ± 42.73 ***	295.62 ± 38.92 **	309.78 ± 50.19 **	318.06 ± 50.64 ***
Calcium in fecal (mg/g dry mass)	46.38 ± 9.50	40.92 ± 10.50	39.32 ± 7.80	46.65 ± 15.30	44.23 ± 16.74	48.29 ± 0.59
PYD (ng/L)	71.41 ± 14.69	75.69 ± 4.42	84.64 ± 9.61	95.47 ± 9.11 ***	91.72 ± 10.48 *	89.65 ± 13.77 *
DPD (ng/mL)	53.59 ± 11.88	59.08 ± 7.25	62.57 ± 8.25	56.36 ± 1.51	61.54 ± 8.83	73.36 ± 5.11 **
CTX (ng/mL)	79.16 ± 3.38	85.46 ± 2.95	87.43 ± 4.81	93.68 ± 8.88 *	90.58 ± 10.29 *	89.78 ± 6.27 *
BALP(ng/mL)	37.56 ± 5.35	40.74 ± 2.51	47.34 ± 8.11 **	40.15 ± 2.20	45.59 ± 8.04	46.36 ± 6.40
OC (pg/mL)	214.54 ± 38.11	203.18 ± 8.53	224.81 ± 20.95	283.10 ± 39.76 ***	275.44 ± 42.84 ***	306.97 ± 43.15 ***
PINP (ng/mL)	8.13 ± 1.30	8.52 ± 1.53	8.54 ± 0.80	11.35 ± 1.27 ***	12.04 ± 1.51 ***	12.80 ± 1.45 ***

PYD: pyridinoline; DPD: deoxypyridinoline; CTX: *C*-telopeptide of type I collagen; BALP: Bone Alkaline Phosphatase; OC: Osteocalcin; PINP: Procollagen Type I *N*-Terminal Propeptide. S: sham rats fed AIN 93M; O: ovariectomized rats fed AIN 93M; OB: ovariectomized rats fed AIN 93M with bisphosphonate; OL: ovariectomized rats fed AIN 93M with probiotic; ODL: ovariectomized rats fed AIN 93M with daidzein and probiotic; OTL: ovariectomized rats fed AIN 93M with tempeh and probiotic. Results of ANOVA analysis followed by Tukey’s post hoc honestly significant difference test showing significant differences between types of diet. Data are presented as mean ± standard deviation. * indicates statistically significant difference (*p* < 0.05) compared to the control group (S). ** indicates statistically significant difference (*p* < 0.05) compared to the control group (O). *** indicates statistically significant difference (*p* < 0.05) compared to both control groups (S & O).

## Data Availability

All data generated or analyzed during this study are available from the corresponding author on reasonable request.

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
