# Peer review of "Impact of *Lactobacillus acidophilus* and Its Combination with Isoflavone Products on Calcium Status, Calcium Transporters, and Bone Metabolism Biomarkers in a Post-Menopausal Osteoporotic Rat Model"

_nutrients, 2024, doi:10.3390/nu16152524_

Round 1

Reviewer 1 Report

Comments and Suggestions for Authors

The authors stated that the purpose of the study was to evaluate the effects of the combination of the probiotic L. acidophilus and isoflavones on calcium status, calcium transporter expression, and bone metabolic biomarkers in a rat model of postmenopausal osteoporosis. The results and experimental contents of the study are sufficient to determine the purpose of the study. However, various hematological parameters were also measured and interpreted as meaningful results. Therefore, these should be mentioned in the purpose of the study in the Introduction. Also, please mention the limitations of the study in the Discussion. The detailed revisions are included in the attached file. 

Comments on the Quality of English Language

Overall, the English is fine. Minor editing of English language required. 

Author Response

Dear Reviewer,

We appreciate your time and attention to our manuscript titled “Impact of Lactobacillus acidophilus and its Combination with Isoflavone Products on Calcium Status, Calcium Transporters, and Bone Metabolism Biomarkers in Postmenopausal Osteoporotic Rats Model,” submitted to Nutrients journal. We acknowledge your review and thank you for considering our work. Thank you for your commitment to the peer-review process. Please find the detailed responses in the attached file and the corrections highlighted in the re-submitted files.

Regards,
Authors

Reviewer 2 Report

Comments and Suggestions for Authors

1. why  L. acidophilus DSM20079 was selected?Does any strain of bacteria have this function?

2. Since the experiment did not set up Daidzein or Tempeh group, how can we determine if the effect is a combination rather than Tempeh or Daidzein?

3. The method and animal grouping sections are too complex and lengthy to understand.

4.Why not directly replace Daidzein or Tempeh with isoflavone, as the latter contains multiple substances. Are there other substances at work instead of isoflavone?

Comments on the Quality of English Language

Quite puzzling.

Author Response

(The authors gave the same response as above.)
